# Epidemiological and Bacteriological Investigations Using Whole-Genome Sequencing in a Recurrent Outbreak of Pullorum Disease on a Quail Farm in France

**DOI:** 10.3390/ani11010029

**Published:** 2020-12-26

**Authors:** Sophie Le Bouquin, Laetitia Bonifait, Amandine Thépault, Thomas Ledein, François Guillon, Sandra Rouxel, Rozenn Souillard, Marianne Chemaly

**Affiliations:** 1ANSES, Epidemiology, Health and Welfare Unit, BP 53, 22440 Ploufragan, France; rozenn.souillard@anses.fr; 2ANSES, Hygiene and Quality of Poultry and Pig Products Unit, 22440 Ploufragan, France; laetitia.bonifait@anses.fr (L.B.); amandine.thepault@anses.fr (A.T.); sandra.rouxel@anses.fr (S.R.); Marianne.chemaly@anses.fr (M.C.); 3SELARL AMI-VET, 57 rue Paul Painlevé, 35150 Janzé, France; t.ledein@ami-vet.fr; 4DGAL, SDSPA/BSA, 251 rue de Vaugirard, 75732 Paris CEDEX 15, France; francois.guillon@agriculture.gouv.fr

**Keywords:** epidemiological investigation, *Salmonella*, pullorum disease, quail, whole genome sequencing

## Abstract

**Simple Summary:**

Although pullorum disease is endemic in many parts of the world, this avian disease responsible for high economic and commercial losses has been eliminated from organized poultry production in Europe and North America. However, it may still remain in backyards and reappear sporadically on conventional poultry farms. This study presents in detail a recurrent outbreak of pullorum disease on a quail farm. In this case report, we present how epidemiological and bacteriological investigations using molecular sequencing tools were carried out in order to identify the source of contamination. Finally, we identify high-risk sanitary practices, and propose recommendations to manage and control this poultry disease, which has become rare in European countries. Given the development of outdoor farms and the increase in self-consumption family farms, a resurgence of pullorum disease cannot be excluded in the coming years. It is essential to develop effective sanitary barriers to prevent transmission between the two coexisting populations of commercial and non-commercial poultry and to raise awareness among all those involved in the poultry industry to be able to detect any outbreak quickly.

**Abstract:**

An outbreak of pullorum disease causing septicemia and high mortality was diagnosed in 2019 on a quail farm in western France. An initial episode had been detected in another building at the same site eight months earlier. Given the exceptional nature and the extent of the potential economic consequences of pullorum disease, epidemiological and bacteriological investigations using molecular sequencing tools were carried out. *Salmonella* Gallinarum and *Salmonella* Infantis were isolated (using the NF U 47-101 reference method) from samples taken from birds at the infected site. A resurgence of the initial episode by horizontal transmission of *S.* Gallinarum is the most likely hypothesis, supported by whole-genome sequencing (WGS) of the strains isolated during the two episodes. Risk health practices have been identified, including the rearing of animals of different ages and species on the same site. Recurrence is explained by the probable persistence of reservoirs of the pathogen on the site (manure, lesser mealworm beetles). The article also highlights the importance of decontamination measures, including pest control, as a key element in the success of the disease control protocol.

## 1. Introduction

Pullorum disease is a septicemic bacterial disease caused by *Salmonella enterica* subsp. *enterica* serovar Gallinarum biovar Pullorum affecting different avian species [1]. This disease is distinct from fowl typhoid, a septicemic disease of adult poultry caused by *Salmonella* Gallinarum biovar Gallinarum (*Salmonella* Gallinarum). Chickens and turkeys are particularly affected, but other species such as quails, ducks, pheasants, partridges, and guinea fowls are not exempt [2,3]. In its acute form, Pullorum disease is almost exclusively a septicemic disease of young chickens causing serious general damage associated with mortality of 50% to up to 100%. However, the organism may also be associated with disease in other avian production and may be carried subclinically or lead to reduced egg production and hatchability in older birds. Game birds and backyard poultry flocks may act as reservoirs of infection and are important in the epidemiology of the disease [1]. Vertical transmission via egg contamination is the major route of transmission but horizontal transmission is also possible [1,2,3].

Although pullorum disease does not have a significant public health impact, it is especially important in animal health and for economic reasons including trade restrictions [2]. Pullorum disease is endemic in many parts of the world (Middle East, Africa, Asia, Central, and South America), where it still causes considerable economic losses [4,5]. The disease has been eliminated from organized poultry production in Europe and North America [5,6] and many countries are reported to be free [6]. Few cases have been identified in Europe over the last decade, notably in Hungary, Sweden, Greece, Netherlands, Serbia, and UK and more recently in Russia and Denmark.

The implementation of a drastic eradication policy, carried out in the 1970s, led to the elimination of pullorum disease in France. However, it may still remain in backyards and reappear sporadically on conventional poultry farms. The last outbreaks observed in France occurred in 1984 and 1985 in laying hens, 2003 and 2004 in guinea fowl, and more recently in 2011 in *Gallus gallus*. Three outbreaks of pullorum disease (*S.* Gallinarum) were identified in different production sectors (broilers, laying hens, and breeders) at that time in northwestern France [7]. The origin of the infection could not be formally identified but an epidemiological link between the farms could be shown, confirmed by the molecular profiles of *S*. Gallinarum strains isolated using the pulsed-field gel electrophoresis (PFGE) method.

In July 2019, a new outbreak was reported on a broiler quail farm in western France. The farm includes four separate rearing sites, with different poultry productions (breeding, broiler and laying quails, chicken broilers, ornamental poultry, pheasants, and partridges). The farmer has his own quail hatchery and slaughterhouse for personal production of chickens and quails. Most of the production is marketed via short circuits, on the farm and at local markets, with the exception of game production, which is sold for export by a production company.

This case occurred following a previous case that had been reported eight months earlier (2018) in another building on the same farm. As this farm was located in the heart of a major free-range poultry production area, intensive epidemiological investigations were decided in close collaboration with the veterinarian, the veterinary officer, and the farmer. The objectives were to prevent the risk of spread to neighboring farms, and to propose effective sanitation measures.

Considering the low number of available epidemiological descriptions regarding pullorum disease, this case report study documents the identification, testing, and epidemiological investigations that were carried out. To our knowledge, this is the first case report using whole-genome sequencing, allowing a better discrimination of the strains, while the latest publications were supported by pulsed-field gel electrophoresis (PFGE) methods [7,8,9]. We also identify high-risk sanitary practices, and propose recommendations to manage and control this poultry disease, which has become rare in European countries.

## 2. Materials and Methods

### 2.1. Case Description

The case occurred on one of the four sites, equipped with two buildings containing exclusively broiler quails. Both buildings are separated into two rooms, each with a capacity of 12,000 animals of the same age. Only one of the two buildings was affected. Mortality started in one of the rooms in 8-day-old animals (cumulative mortality of 16% over 2 weeks). The animals in the other room were slightly older (15 days) and were also affected, but later and to a lesser extent.

At necropsy, septicemic lesions were found by the veterinarian and organs were removed for bacteriological analysis. Blood samples were also taken for serological analysis. Antimicrobial treatment (trimethoprim-sulfadiazine) was immediately started. At the same time, a declaration of suspicion was communicated to the veterinary authorities and commercial restrictions were imposed.

An initial outbreak of *Salmonella* pullorum had been discovered at the same site 8 months earlier (November 2018), but in the second building.

### 2.2. Epidemiological Investigations

The farm was the target of a collaborative in-depth epidemiological investigation using a questionnaire covering outbreak description, husbandry practices, biosecurity, animal movements, and possible farm-to-farm contacts. The background of the first *Salmonella* pullorum infection on the farm was documented.

About 10 poultry farms and a few backyards were identified within 3 km and were clinically monitored by their health veterinarian as specified in French regulations [10]. As part of this process, a close monitoring combining clinical surveillance and samples for bacteriological research of two free-range farms (one laying hen farm and one broiler farm) and two backyards located in the immediate vicinity of the infected farm was also carried out, as epidemiological links by geographical proximity could not be ruled out.

Close monitoring of the cleaning and disinfection operations following the depopulation, and a check on their effectiveness were carried out before restarting the farming activity at the site.

### 2.3. Sampling Protocol

Samples were taken by the veterinarian and the competent veterinary authorities at the three key stages of monitoring: (i) during clinical suspicion in the affected animals, (ii) as part of the epidemiological investigation, and (iii) during the cleaning and disinfection operations. Sampling was carried out at different places of interest: at the infected site to confirm the clinical diagnosis, at other farm sites to monitor the health status of other animals kept on the farm (in particular breeding quails and game), and on epidemiologically linked farms to check for possible spread to other production facilities. Several types of samples were sent to the laboratory for analysis: organs from dead animals (hearts, livers, caeca) for bacteriological testing, as well as blood for specific antibody detection by rapid slide agglutination (RSA), environmental swabs, and lesser mealworm (*Alphitobius diaperinus*) beetles. A summary of samples taken is presented in Table 1.

### 2.4. Bacteriological Analysis

In this national study, dead animals and organ samples were analyzed by the national reference laboratory for *Salmonella*, according to the French Standard NF U 47-101 [11] (Animal health analysis methods—Isolation and identification of any *Salmonella* serotypes or of specified *Salmonella* serotypes among birds). As recommended by the reference method used in France (NF U 47-101), organs (liver, heart, spleen, and cecum samples) from the quails were diluted, homogenized in 1:10 with Buffered Peptone Water (Biomérieux, Marcy-l’Etoile, France) and incubated for 16 h to 20 h at 37 °C. In parallel, a direct isolation on rich agar medium of the organs presenting the lesions is carried out. Enrichment was done with three media (Muller Kauffmann tetrathionate broth (MKTTn, Biokar, Allonne, France), selenite cystine broth (SC, Oxoid, Dardilly, France), and modified semi-solid Rappaport-Vassiliadis (MSRV, Biokar, Allonne, France)). MKTTn and MSRV were incubated for 24 h at 41.5 °C, whereas SC broth was incubated at 37 °C for 24 h. After enrichment MKTTn suspension were plate on xylose lysine tergitol 4 (XLT4, Biokar, Allonne, France) and MSRV and SC suspension were plate on RAPID’Salmonella (R’S, BioRad, Marnes-La-Coquette, France) and xylose lysine desoxycholate (XLD, Biokar, Allonne, France). The agar plates were incubated for 24 to 48 h at 37 °C. Characteristic colonies were biochemically tested for glucose fermentation, lactose oxidation, hydrogen sulfide, and gas production on Kligler Hajna medium (KH, Biokar, Allonne, France) incubated for 24 h at 37 °C. All *Salmonella* isolates were confirmed by serotyping according to the Kauffmann–White scheme using slide agglutination test [12].

The detection of *Salmonella*-specific antibodies was carried out by official laboratories according to the French Standard NF U 47-034 [13] (Animal health analysis methods—Detection of antibodies specific for *Salmonella* Pullorum Gallinarum in the serum by rapid slide agglutination). This method consisted of a rapid slide agglutination (RSA) test, based on a reaction between half antigens (from standard (O: 1, 9, 121, and 123) and variant (O: 1, 9, 121, and 122)) and half serum (25 µL). The observation of an agglutination led to a positive result.

### 2.5. Molecular Analysis

Genomic DNA of *Salmonella* strains was extracted from one-day single-colony cultures using a QIAamp DNA Mini Kit (QIAGEN, Marseille, France), and quantified using a Qubit 2.0 fluorometer and the Qubit dsDNA (double-stranded DNA) HS (high-sensitivity) assay kit (Thermo Fisher Scientific, Saint Herblain, France). All *S*. Gallinarum strains were sequenced using Illumina technology by the Paris Brain Institute (Institut du Cerveau et de la Moelle épinière, ICM, Pitié-Salpêtrière Hospital, Paris; www.icm-institute.org). Libraries were prepared using the Nextera XT DNA Library Preparation Kit and Nextera XT Index Kit (96 indexes) (Illumina). Samples were then sequenced with a NextSeq 500 machine using the NextSeq 500 Mid Output Kit v2 (300 cycles) (Illumina, Evry, France). Paired-end raw reads were deposited on the public EnteroBase database platform for *Salmonella* (http://enterobase.warwick.ac.uk/), and were automatically de novo assembled using SPAdes [14] once the sequences had been uploaded. Further details on assembly pipelines are available on the EnteroBase website (https://enterobase.readthedocs.io/en/latest/pipelines/enterobase-pipelines.html). The genomic comparison of strains was carried out using the cgMLST scheme available on EnteroBase including 3002 genes [15]. Similarly to 7-gene mutlilocus sequence typing (MLST), a specific sequence type (ST) is attributed to each unique allelic combination, based on 3002 loci, hereafter defined as cgSTs. Similar but non-identical strains (strains showing different cgSTs) were identified in EnteroBase by using the hierarchical clustering method (HierCC) that allows clustering of strains which can differ up to a specified and fixed number of cgMLST alleles [15].

Due to the rarity of the serovar Gallinarum in France, 28 strain assemblies from international European countries were included in the comparison to assess the genetic similarity of the strains (Appendix A). A neighbor-joining tree was created in EnteroBase using GrapeTree [16] and the RapidNJ algorithm [17]. Assemblies are publicly available from EnteroBase (S18LNRS01-09: enterobase barcode SAL_ZA0109AA; S19LNRS08-01: enterobase barcode SAL_ZA0111AA).

## 3. Results

### 3.1. Epidemiological Investigations and Follow-Up of Cleaning and Disinfection Effectiveness

The simultaneous presence on the same farm of many poultry species and several ages on the same site (multi-age and multi-species farming) have been identified as major risk factors for the maintenance of infection on a farm. Under these conditions, crossing circuits of animals, equipment, or personnel is unavoidable.

The epidemiological investigation also provided information on the background to the first outbreak, including the management of the cleaning and disinfection operations following the slaughter of the birds. Assessment of these operations proved favorable and the site was authorized to resume activity at the beginning of 2019. The manure from this first outbreak had been limed, removed, and stored in piles near the farm premises for several months. The farmer did not observe any combustion of this manure pile. In early spring 2019, it was then picked up by another farmer for spreading on fields.

Concerning the investigation of epidemiological links, as quail production is conducted in self-sufficiency, forward and backward tracing showed limited contacts. None were considered relevant to investigate during this period. Investigation of neighborhood links and geographically nearby farms did not make it possible to detect any clinical cases. All the bacteriological and serological analyses carried out during the investigations on the farms geographically linked to the outbreak were negative.

As soon as the infected flock was eliminated, the site was immediately subjected to a deep cleaning and disinfection protocol. An assessment of the effectiveness of the cleaning and disinfection was carried out at the end of the process. During the visit, the premises were considered visually clean. However, the persistence of lesser mealworm beetles, in particular in the sanitary airlock where staff change clothes, wash, and disinfect their hands, was noted. The farmer reported that he was particularly concerned about the recurrence of these beetles at his premises.

### 3.2. Bacteriological Results

In 2018, the isolated strain of S. Gallinarum (S18LNRS01-09) sent to the French National Reference Laboratory (NRL) for Salmonella was confirmed by serotyping analysis (formula: 1,9,12:-:-). The diagnosis of pullorum disease was based on the isolation of Salmonella Gallinarum (*Salmonella enterica* subsp. enterica serovar Gallinarum) from samples taken from dead animals at the suspected site. In 2019, the NRL for Salmonella established the quail’s infection, by S. Gallinarum and *Salmonella* Infantis, by the detection of the strains in the heart and liver (strain S19LNRS08-01). *Salmonella* Infantis, a *Salmonella* that had remained on the farm for several years and was previously detected especially during the previous case, was also identified.

The detection of specific antibodies by rapid slide agglutination (RSA) revealed five positive sera (5/60) in the birds from the affected building.

All bacteriological and serological tests carried out in animals from other poultry productions on the farm were negative. Environmental samples taken at the infected site were positive for S. Infantis, but all were negative for S. Gallinarum.

The environmental swabs taken after the cleaning and disinfection of the premises revealed the presence of S. Infantis on the floor and walls. Salmonella Infantis was also found on lesser mealworm beetles still present in the room. Salmonella Gallinarum was not found. A summary of the analytical results is presented in Table 1.

### 3.3. Whole-Genome Sequencing Results

Fifty-nine and fifty-eight contigs were obtained for the S. Gallinarum sequenced genomes from 2018 and 2019, respectively. The total assembled sequence length was 4,781,993 bp for strain S18LNRS01-09 and 4,776,421 bp for strain S19LNRS08-01. To compare the genome of the two strains of S. Gallinarum, the core-genome MLST approach was used (cgMLST scheme available on the EnteroBase website). The S18LNRS01-09 and S19LNRS08-01 strains belonged to cgST199405 and cgST199407, respectively (Table 2).

Hierarchical clustering of cgMLST (HierCC) defines clusters based on cgMLST. The distances between genomes were calculated using the number of shared cgMLST alleles. Both strains have a different core-genome sequence type but they are grouped in the same hierarchical cluster (HC) using 10 different cgMLST allele distances. The comparison of their allelic profiles on 2989 loci highlighted six differences: allelic variations on five loci between strains, and one locus was missing or truncated in one strain while present in the second strain. Therefore, these strains seem to be phylogenetically closely related and may be considered (according to the European Union Reference Laboratory (EURL) for Salmonella recommendation) to have a common source of contamination. These results were emphasized by the gene-by-gene comparison of strains from France with all the international European S. Gallinarum strains available from the EnteroBase data collection. Importantly, while strains from France harbored five allelic variations between each other, they showed significant differences with international European strains (>200 allelic variations) (Figure 1).

Strains are colored on the NJtree according to their country. Clusters generated using the hierarchical clustering method from EnteroBase and using a 200 cgMLST allele distance (HC200) are represented by circles. *S.* Gallinarum strains from France sequenced in this study showed five allelic differences between them, while more than 200 differences were observed between French and other European strains.

## 4. Discussion

The epidemiological investigation did not enable us to identify retrospectively the origin of the contamination that occurred in November 2018. A major issue was to determine whether this second outbreak was a new introduction or a resurgence of the disease. Resurgence is defined as the reappearance of the disease on a previously infected and subsequently sanitized farm, without further introduction of the pathogen [18]. Several epidemiological findings support this hypothesis.

The first relates to the maintenance at the site of manure previously contaminated during the first outbreak, which may have been a source of persistence of infection. Despite lime application, the farmer did not see any reduction in the volume of the pile or any real combustion of the manure. The resistance of *Salmonella* Gallinarum biovar Pullorum is similar to other *Salmonella* bacteria. Incineration or composting ensuring a rise in temperature to 65 °C in the center (55 °C on the surface) for several days is required to obtain their destruction and secure the effluents [19,20,21,22]. Probably insufficiently humidified, the manure would not have risen sufficiently in temperature to allow its sanitization. As a result, run-off during the winter period as well as the removal operation of the contaminated manure pile in early spring could have been the cause of spread of *Salmonella* still present.

The presence of a large number of lesser mealworm beetles was detected on this farm. The observation of these beetles inside the premises and in the airlock after the desensitization, rodent removal, cleaning, and disinfection operations alerted the veterinarian. The hypothesis of a possible persistence of infection via the beetles was reinforced by the results of environmental samples taken after cleaning and disinfection on which *S.* Infantis was detected in areas usually easy to clean (walls and floor of the room). The collected beetles also tested positive for *S*. Infantis. Lesser mealworm beetles are known to be reservoirs and mechanical vectors for many pathogens, including zoonotic agents such as *Salmonella* [23]. Under experimental conditions, *Salmonella* can be detected in their larvae for up to 7 days from a contaminated substrate [24]. In such cases, transmission of the infection by ingestion is the most likely pathway. Other passive vectors are known to transmit *S.* Pullorum, including red mites [1] and rodents. For example, Anderson et al. [8] reported the presence of an *S.* Pullorum strain in the intestine of a rat trapped in an outbreak, which was very similar to the strain identified in poultry. In our case, although *S.* Gallinarum could not be isolated directly on the lesser mealworm beetles, the detection of *S*. Infantis is a good indicator of the maintenance of *Salmonella* contamination in this facility.

Finally, whole-genome sequencing allows better discrimination than pulsed-field gel electrophoresis (PFGE) usually used as the gold standard method for *Salmonella* investigation. In this study, WGS results provided strong evidence for the epidemiological link between the strains in the two episodes (a single cluster at HC10) and confirmed the recurrence hypothesis of the epidemiological investigation. These tools allow for a more comprehensive investigation to identify the strains linked to each other, and thus confirm or refute the existence of an epidemiological link between outbreaks. No epidemiological link could be demonstrated with other European strains of *S.* Gallinarum (Pullorum) sequenced in EnteroBase, supporting the hypothesis of a resurgence in the quails farm.

Comprehensive investigations were carried out in the breeding quails reared at another site. Transmission of pullorum disease can result from vertical or horizontal contamination, but vertical transmission through eggs due to contamination of the ova is the most common mode of transmission [2], or indirectly through contact from chick to chick in the hatchery. Direct or indirect horizontal transmission appears to be epidemiologically less common, but it is also described in particular between epidemiologically linked farms, via litter, water or feed, animal movements [9], and rodents [8]. In our present case, all the samples taken from the breeders were negative, focusing on horizontal contamination, as manure and lesser mealworm beetles may have acted as a reservoir and ensured the maintenance and spread of the bacteria at the site.

Evidence from countries where the disease remains enzootic shows that multi-age livestock production is an important factor for the persistence and spread of the disease. Non-commercial farms are often cited as potential reservoirs of the disease [8,25]. Since these farms are poorly monitored from a health point of view, they may represent a greater risk because *Salmonella* often circulates there subclinically, without any signs being detected in the birds. According to several authors, many cases are likely to occur in backyard flocks and disease occurrences are under-reported in official data. Therefore, the prevalence of pullorum disease is probably underestimated [6,9]. It is thus important to detect and eliminate the disease from backyard premises in order to prevent its introduction on commercial farms.

In this context, backyards identified near the affected site were monitored with clinical and sampling visits in order to check their status and rule out their potential role as a reservoir of the disease.

*S.* Gallinarum was not isolated from any of the environmental samples. In contrast to classical (motile and H2S+) *Salmonella*, which can be detected in the environment using swabs, dust, or droppings samples, it is really difficult to detect *S.* Gallinarum in this way [26]. The method of selenite-cystine enrichment at 37 °C, which can be used to detect *S.* Gallinarum, is not very sensitive and not particularly selective for environmental samples. However, the identification of *S.* Infantis on a recurrent basis for several years and during the control carried out following cleaning and disinfection is a good indicator of the possible maintenance of *Salmonella*-like pathogens on the farm.

## 5. Conclusions

The detection of an outbreak of *Salmonella* disease on a commercial farm is an event that has now become exceptional in France. Cases observed are rare and not recent. However, the development of outdoor farms and the increase in self-consumption family farms are likely to lead to a recrudescence of this disease, as well as other diseases that are currently considered historical. Therefore, it is essential to raise awareness among all those involved in the poultry industry (farmers, health and technical staff, veterinarians, and diagnostic laboratories) to be able to detect any outbreak quickly.

The co-existence of many poultry species and multi-ages on the same farm have been identified as major risk practices for the persistence of infection on the farm. The management of cleaning and disinfection operations is a key element of the control measures. In our case, the implementation of an additional decontamination step and a reinforced pest control protocol were necessary to eradicate the disease.

## Figures and Tables

**Figure 1 animals-11-00029-f001:**
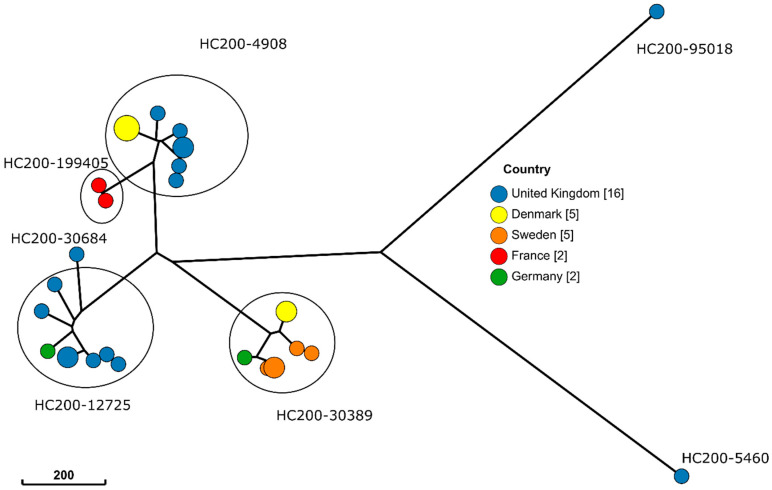
Neighbor-joining tree (NJTree) of European *Salmonella* Gallinarum strains, available from the EnteroBase data collection, compared using the EnteroBase cgMLST scheme.

**Table 1 animals-11-00029-t001:** Summary of samples taken and serological and bacteriological results obtained.

Location	Type of Samples	Results
Suspected site - on clinically suspect quail	Autopsies Bacteriology (heart, liver, caeca, *etc*.) Serology (60 blood samples/building)	Septicaemia *Salmonella* Infantis *Salmonella* Gallinarum 55 RSA neg, 5 RSA pos
Other poultry present on the farm - breeding quails	Autopsies Serology (2 × 60 blood samples taken 10 days apart)	RSA neg
Poultry farms in the vicinity - backyards - laying hen farm - broiler chicken farm	Serology (4 to 60 blood samples/building)	RSA neg
At the infected site after cleaning and disinfection operations	swabs lesser mealworm beetles	*Salmonella* Infantis *Salmonella* Infantis

**Table 2 animals-11-00029-t002:** Core genome MLST profile of *Salmonella* Gallinarum strains.

cgMLST								
Strain	Serotype	Year	Sector	HC0	HC2	HC5	HC10	HC20
S18LNRS01-09	*Gallinarum*	2018	Quails	199405	199405	199405	**199405**	**199405**
S19LNRS08-01	*Gallinarum*	2019	Quails	199407	199407	199407	**199405**	**199405**

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
