# Peer review of "Epidemiological and Bacteriological Investigations Using Whole-Genome Sequencing in a Recurrent Outbreak of Pullorum Disease on a Quail Farm in France"

_animals, 2020, doi:10.3390/ani11010029_

Round 1

Reviewer 1 Report

The authors extensively revised the manuscript the intro is well written and informative, however, my suggestion is to check the whole text again and remove the grammatical mistakes and also few sentences are bit long so make it short or split it.

Author Response

Thank you for your comments.You will find below our answers.

In the revised manuscript, the modifications are marked using the "Track Changes" function in Microsoft Word.

As you have asked us, we have tried to improve the form of the article, in particular by shortening some sentences (e.g. lines 89-91 and lines 242-245)

Reviewer 2 Report

Dear authors,

Thank you for addressing my concerns. I only have three comments:

  • I suggest changing or removing the third sentence in the abstract at line 30. "A drastic eradication policy implemented in the 1970s eliminated pullorum disease, but it can still reappear sporadically on poultry farms." It is not relevant to the aritcle and is inserted in the mid of the case presentation.
  • In the abstract, at line 37, it should be genome, not genotype.
  • The supplementary table is not refered anywhere in the text.

Best regards.

Author Response

Thank you for your comments.You will find below our answers.

In the revised manuscript, the modifications are marked using the "Track Changes" function in Microsoft Word.

As proposed, the sentence has been deleted from the abstract lines 30-31

We have replaced by whole-genome sequencing

The table has been mentioned in the manuscript (line 204). Thank you for your comment.

Reviewer 3 Report

Although this revised version is an improvement over the original manuscript, I still think the work can be refined significantly. For example, the first two sections of the Materials and Methods still read like introductory text to me, and perhaps they should be moved to the Introduction. Also, some of the Methods require a reference, such as the "French Standard" methods mentioned in lines 158-160 and 175-177. Most readers will be unfamiliar with these methods, and a citation is required for each that shows the specific details associated with them.

Furthermore, in lines 177-179, how was the rapid slide agglutination test performed? Details of the protocol must be included here. If the test was done according to the manufacturer's instructions, this must be stated, with the name of the manufacturer listed.

The Results section is written in such a way that much of it reads like a Discussion section, and therefore I suggest combining these into one section entitled "Results and Discussion." This will likely require some rewording and possibly also reorganization here and there, but I think it will improve the paper and make it feel more like a research article and less like an essay.

Author Response

Thank you for your comments.You will find below our answers.

In the revised manuscript, the modifications are marked using the "Track Changes" function in Microsoft Word.

We have moved the description of the farm (L112-117) to the introductive part (L78-83), but kept the case description in M&M. For our part, we consider that epidemiological investigations are part of the Material and Methods, in the same way as the laboratory methods used.

Two references for French standard methods have been added in the bibliography section

The paragraph regarding rapid slide agglutination test has been rewritten with additional informations as requested

We have considered the suggestion to merge the results part and the discussion part. From our point of view, the results presented in the results section are strictly limited to the observations made in the study. On the contrary, the discussion offers a good perspective of the results in relation to previous knowledge. It seems to us more rigorous to separate these two aspects, so that the reader can easily identify the new knowledge acquired in the study. We had a specific discussion on this point with the Academic Editor who finally proposed to keep the manuscript how it is.

This manuscript is a resubmission of an earlier submission. The following is a list of the peer review reports and author responses from that submission.

Round 1

Reviewer 1 Report

  1. In abstract please write the key results and remove the detail methodology which is used in this study. Focus on the results and also the conclusion of your work.
  2. The scope of the work is not clearly described while the main problem is the arrangement of introduction. Re-organize the introduction section by connecting the previous paragraph to make it fluent for the readers. Additionally, the intro is too short and some paragraphs are also short. Rewrite it and add sufficient information’s with their latest references.
  3. There is a big difference in the essay writing and scientific writing. The methodology is really very poor and not look like a scientific paper methodology. Only one reference is cited in the whole section, which techniques were used? Follow some latest papers for writing this section. Re write it again by adding the latest references, technology name, equipment names along their manufacture etc.
  4. The quality of the English language of the manuscript, that I believe could and should be improved. Although the text is understandable, there are some grammar mistakes, and unclear sentences; the manuscript should be carefully checked and corrected, best by a professional service for editing academic texts in English or native speaker.
  5. The scientific names should be italic, check whole text.

Reviewer 2 Report

Authors present a case report of a recurrent outbreak of S. Gallinarum in a french broiler quail farm with an epidemiology follow-up where bacterial test and WGS of both events were compared.

I concur that this case report is, as the autors claim, important since robust approaches to outbreaks are more effective in slowing down such bacterial diseases, thus improving the economy of the sector.

I have found, however, some issues that should be addressed:

  • WGS results (i.e. either the sequencing reads or the assembled sequence, preferably both) need to be publicly accesible so the paper results may be corroborated independently.
  • Include the link and reference of the EURL for Salmonella where the Lab discusses the common sources of contamination and strains simmilarity degrees.
  • List the strains used for the phylogenetic tree.
  • Bettle discussion pharagraph (lines 269-282) has no conclusion, I suggest the authors revise it.
  • My main concern is the last conclusion. The first part (lines 327-329) is tenuously sustained by the mention of the slaughter from the first oubreak, however no economic impact analysis was offered. The second part (lines 330-332) cannot be drawn from the rest of the paper.

Reviewer 3 Report

Comments to the Authors

The manuscript entitled “Epidemiological and bacteriological investigations using whole-genome sequencing in a recurrent outbreak of pullorum disease on a quail farm in France” (Manuscript ID: animals-927979) describes a molecular (whole-genome sequencing) epidemiological study conducted on a quail farm in France that had endured multiple recent outbreaks of pullorum disease. Although rare, outbreaks of this disease have the potential to cause significant economic impact and can teach useful lessons regarding effective sanitation protocols for both commercial and noncommercial poultry operations.

This journal (Animals) is perhaps not a bad choice for this work, but I would think that a journal more focused on pathogen transmission would be more appropriate, such as the new MDPI journal Epidemiologia. Possibly also Veterinary Sciences would be a good choice if it is animal husbandry that the authors wish to emphasize. If it is the causative agent (Salmonella enterica serovar Gallinarum) itself that is the focus, perhaps the journal Pathogens or even Microorganisms would work better. It just seems like there are several other choices, even among MDPI journals alone, that might be more suitable for this work, even if it is not totally out of context for Animals.

This paper is straightforward and does not offer any novel or especially insightful conclusions. However, it is scientifically sound and has merit as a case study report with recommendations as to best practices for containment and the prevention of disease resurgence. Perhaps its main strength is that it may serve to increase awareness of this rare (and rarely considered) but nevertheless important disease that has the potential to cause significant epidemiological and economic hardship.

Although the manuscript is generally well written, I note a few specific comments below that should be addressed before it is suitable for publication.

Comments:

1)    Abstract, Line 16: Probably the authors mean to say “for” instead of “of” high economic and commercial losses. Also, the comma is unnecessary in this line.

2)    Keywords, line 43: I recommend spelling out “WGS” fully since the authors are listing this as a searchable keyword.

3)    Line 46: My understanding is that “Salmonella enterica sp. enterica” should be “Salmonella enterica subsp. enterica.” The first “enterica” denotes the species epithet, whereas the second would denote the subspecies.

4)    Line 50: Why not simply say “...mortality of up to 100%” instead of “...mortality of up to 50 to 100%”? Or perhaps word it as “...mortality of 50 to up to 100%.” The way it is written is awkward.

5)    The word “very” is used in several places throughout the manuscript (see lines 40, 42, 52, 182, 282, and 315, for example). In nearly every case, the word can be deleted without affecting the reading of the text at all. Therefore, it is unnecessary and can even be distracting if overused. I recommend removing the word in every case in which it adds essentially nothing to the text, which is nearly every time. Perhaps it is helpful in line 316, but even here, it could be replaced with a better word, such as “especially.”

6)    Line 100: I recommend listing the drug in parentheses as “(TMP-SMZ, sulfamethoprim)” or perhaps “(TMP-SMZ, trimethoprim-sulfamethoxazole)”.

7)    Line 113: Does “close monitoring” mean visual monitoring or molecular monitoring of samples? Please clarify.

8)    In seems to me that Table 1 should in the Results section rather than the Materials and Methods section since results are listed among the information included. Also in this table, I’m guessing it should be “Serology (4´60 blood samples/building)” rather than “Serology (4-60 blood samples/building)”?

9)    Lines 155-156: Please define the “MLST” abbreviation. I don’t think this is done elsewhere in the manuscript.

10)  Line 190: Please briefly describe the “airlock” for unfamiliar readers.

11)  Lines 241-242: Considering that France is in Europe, would the “European strains” be better described as “international European strains”?

12)  Several places in the Discussion (e.g., lines 268, 277, and 278), the genus Salmonella is not capitalized. Please correct.

13)  Line 314: The word should be “motile” not “mobile.”
